# Exploring TREC and KREC Levels in Nursing Home Residents and Staff and Their Association with SARS-CoV-2 Antibody Response After Vaccination

**DOI:** 10.3390/vaccines13080874

**Published:** 2025-08-19

**Authors:** Eline Meyers, Natalja Van Biesen, Liselore De Rop, Tine De Burghgraeve, Marina Digregorio, Laëtitia Buret, Samuel Coenen, Beatrice Scholtes, Jan Y. Verbakel, Stefan Heytens, Piet Cools

**Affiliations:** 1Department of Diagnostic Sciences, Faculty of Medicine and Health Sciences, Ghent University, 9000 Ghent, Belgium; eline.meyers@ugent.be (E.M.);; 2LUHTAR, Leuven Unit for HTA Research, Department of Public Health and Primary Care, KU Leuven, 3000 Leuven, Belgium; 3Research Unit of Primary Care and Health, Department of General Medicine, Faculty of Medicine, University of Liège, 4000 Liège, Belgium; 4Department of Family Medicine & Population Health, Faculty of Medicine and Health Sciences, University of Antwerp, 2000 Antwerp, Belgium; samuel.coenen@uantwerpen.be; 5NIHR Community Healthcare Medtech and IVD Cooperative, Nuffield Department of Primary Care Health Sciences, University of Oxford, Oxford OX1 2JD, UK; 6Department of Public Health and Primary Care, Faculty of Medicine and Health Sciences, Ghent University, 9000 Ghent, Belgium

**Keywords:** TREC, KREC, SARS-CoV-2 antibodies, COVID-19 vaccination, nursing home residents

## Abstract

Background: T-cell receptor excision circles (TRECs) and kappa-deleting recombination excision circles (KRECs) are markers of recent thymic and bone marrow output, respectively. As they have previously been associated with immunosenescence, we aimed to investigate their association with anti-spike SARS-CoV-2 (S1RBD) IgG antibody response after COVID-19 vaccination in nursing home residents (NHRs) and staff (NHS). Methods: We measured TREC and KREC levels and S1RBD IgG antibody levels from dried blood spots (DBSs) using in-house qPCRs and a commercial ELISA kit, respectively, in 200 participants (50 NHRs and 150 NHS). DBSs were collected in April 2021, approximately two months after primary course COVID-19 vaccination (BNT162b2). We assessed the association between TREC and KREC as dependent variables and age, sex, infection-priming status, and post-vaccination S1RBD-specific IgG concentrations as independent variables by simple and multiple linear regression. Results: TREC and KREC levels were significantly lower in NHRs compared with NHS and were negatively correlated with age (*p* < 0.001). Neither TREC nor KREC levels were significantly associated with SARS-CoV-2 antibody concentrations (*p* > 0.05). Conclusions: In our study population, TREC and KREC levels decreased with age and were statistically significantly lower in NHRs than NHS. They were, however, not associated with the antibody response after COVID-19 vaccination. Yet, additional research is warranted to explore their potential relevance in cellular immune responses or in combination with other biomarkers of immune function.

## 1. Background

T-cell receptor excision circles (TRECs) and kappa-deleting recombination excision circles (KRECs) are established markers of recent thymic and bone marrow output, respectively. TRECs serve as indicators of naïve T-cell neogenesis, while KRECs serve as indicators for naïve B-cell neogenesis [1]. Both are episomal DNA fragments generated during V(D)J recombination, which is the random rearrangement of gene segments during lymphocyte maturation to assemble a diverse and unique repertoire of T- and B-cell receptors. A visual overview of the formation of TRECs and KRECs is given in Figure 1. Specifically, during T-cell maturation, lymphoid progenitors migrate to the thymus, where the α-chain of a T-cell receptor undergoes VJ recombination. Hereby, the *TCRD* gene, which is located within the *TCRA* locus on chromosome 14, is excised. This excised DNA forms an episomal DNA circle that is ligated at its signal joint DNA region, being a signal joint TREC (sjTREC) [2,3].

KRECs, on the other hand, are formed during the maturation of progenitor B-cells in the bone marrow. KRECs are generated in B lymphocytes that, after the successful VDJ recombination of the immunoglobulin heavy chain, fail to produce a functional VJ recombination of the *immunoglobulin kappa (IGK) light chain* genes on one or both alleles. In such cases, the *IGK* locus is inactivated through a secondary recombination event involving the *kappa-deleting element (κde)*. This element recombines with a *recombination signal sequence (RSS)* located in the intron between the *Joining Kappa (Jκ)* and *Constant Kappa (Cκ)* gene segments. As a result, part of the *IGK* locus is deleted, and the excised DNA forms a circular episomal structure known as a kappa-deleting recombination excision circle (KREC) [1,3].

Both TRECs and KRECs do not replicate during cell division as they are not part of the genomic DNA, making their abundance diluted during peripheral lymphocyte proliferation [3].

TRECs and KRECs are widely used as markers in newborn screening programmes to detect severe combined immunodeficiency (SCID) and X-linked agammaglobulinemia, respectively. As undetectable or low levels of TRECs or KRECs demonstrate defective lymphocyte development, they aid in the early diagnosis of these immune disorders [3]. In many countries, TREC and KREC assays have already been integrated into national routine screening programmes, while in others, pilot studies are being conducted to evaluate their feasibility and effectiveness for broader implementation [4,5,6,7,8]. The global adoption of TREC/KREC screening reflects the growing recognition of these markers. Moreover, TREC and KREC levels can be detected in dried blood spots (DBSs) and are therefore well-suited for application in newborns due to the ease of collection and minimal invasiveness.

Apart from the use of these markers in paediatric settings, they have also been investigated in adult populations, yet for research purposes. Several studies in persons ranging from 1 month to 74 years old have demonstrated that TREC levels are inversely correlated with age [9,10,11]. Moreover, a number of studies have suggested TREC as a marker of immunosenescence [11,12,13,14], as both are characterised by thymic involution and a decrease in naïve T-cells [15]. Specifically, Mitchell et al. investigated TREC levels in 58 to 104 year olds and suggested that immunosenescence is not merely a measurement of chronological age, as immune exhaustion can occur at different ages [12]. In contrast, evidence linking KRECs to immunosenescence is not prominent. Yet, haematopoiesis in the bone marrow is also affected by ageing [16], and it could therefore be hypothesised that KRECs are similarly linked to the deterioration of immune function observed in older adults.

In this context, TREC and KREC levels may hold additional value as predictors of immune function and, more specifically, vaccine response. One study found that TREC levels were positively correlated with the humoral response to influenza vaccination in older adults [17]. In another article, the same authors found TRECs to be associated with a number of genes, epigenetic regulatory elements, and protein markers involved in the immune response to influenza vaccination in older adults [13]. Another study in old macaques demonstrated that macaques with increased TREC levels had higher antibody responses to influenza vaccination [18]. To our knowledge, apart from the studies mentioned, no other studies have been published on the association between TREC or KREC levels and the immune response to vaccination. Alternatively, since the start of the COVID-19 pandemic, a number of studies have demonstrated an association between TREC and/or KREC levels and COVID-19 disease severity, suggesting TRECs and KRECs as biomarkers for COVID-19 disease outcome [19,20,21,22].

Based on the previously discussed evidence, we hypothesised that TREC or KREC levels might be associated with COVID-19 humoral vaccine response. If so, they could serve as biomarkers for humoral vaccine response, providing insights into the mechanisms underlying poor vaccine responsiveness. Therefore, in this study, we explored TREC and KREC levels in a population of nursing home residents (NHRs) and staff (NHS) and their relation to SARS-CoV-2 antibody response after COVID-19 vaccination.

## 2. Methods

### 2.1. Study Design and Population

DBSs from 50 NHRs and 150 NHS were randomly selected from a larger study population of NHRs (n = 500) and NHS (n = 250) [23], collected within a national serosurveillance study in NHRs and NHS residing in Belgium (SCOPE study) [24], using an online tool for random selection (GraphPad Software Inc., San Diego, CA, USA) [25]. These DBSs from NHRs and NHS were collected in April 2021, approximately two months after primary course BNT162b2 vaccination (30 μg, two-dose regimen). The sample size was calculated a priori using GPower software (G Power Version 3.1, Heinrich-Heine-University, Düsseldorf, Germany) [26] for correlation using a bivariate normal model. We assumed a medium effect size (r = 0.3), a significance level (alpha) of 0.05, and a power of 0.8, resulting in a required total sample size of 84. Initially, we included 50 NHRs and 50 NHS. However, during interim analysis, it became evident that a substantial proportion of NHR participants had undetectable TREC levels. To address this and maintain statistical power for our planned analyses, we expanded the NHS group by including an additional 100 participants, who were expected to have measurable TREC levels due to younger age and more active thymic function.

### 2.2. Ethical Considerations

The SCOPE study received approval from the Ethics Committee of Ghent University Hospital (reference number BC-08719) and was conducted following the ethical principles of the Declaration of Helsinki. Informed consent was obtained from all participants or their legal representatives after explaining this study’s aims and procedures.

### 2.3. Questionnaires

Sociodemographic and clinical data were gathered through online questionnaires (LimeSurvey version 3.22, LimeSurvey GmbH, Hamburg, Germany), filled out by the NHS and, in the case of NHRs, by the nursing homes’ head nurses based on their medical files. Collected data included the participant characteristics of age, sex, comorbidities (cardiovascular disease, diabetes mellitus, hypertension, severe renal/lung/cardiac disease, immunodeficiency/immunosuppression, active cancer), and care dependency (for NHRs only) based on the Katz evaluation scale [27]. Moreover, information concerning the COVID-19 vaccination history (dose count, dates, and vaccine brand) and (self-)reported infection history (verified by PCR or antigen testing, or CT scan, along with test results and testing dates for positive cases) was collected. When a SARS-CoV-2-positive test dated ≥14 days before primary course vaccination was reported, the respective participant was classified as infection-primed.

### 2.4. SARS-CoV-2 Antibody Response

To quantify (anti-spike SARS-CoV-2) S1RBD IgG antibodies, the DBS samples were analysed using the SARS-CoV-2 S1RBD IgG ELISA assay (ImmunoDiagnostics Limited, Hong Kong) as previously described [28]. Briefly, a spot of 6 mm diameter was punched from the DBS and placed into 250 µL of S1RBD IgG ELISA buffer. After a one-hour incubation at 37 °C, the extract was diluted 100-fold and transferred to an S1RBD-coated well plate. The remainder of the procedures followed the manufacturer’s ELISA assay instructions, with an optical density (OD) reading at 450 nm using the Behring ELISA Processor III (Siemens AG, Munich, Germany). Samples with an OD higher than that of the highest standard value were retested using 10- and 1000-fold dilutions. Antibody concentrations were expressed in international units/mL (IU/mL) and calculated using a four-parameter logistic (4PL) regression curve (GraphPad Prism version 10.3.0 (GraphPad Software Inc., San Diego, CA, USA) based on a set of SARS-CoV-2 antibody standards provided in the kit. A seropositivity cutoff value of 26 IU/mL was experimentally determined [28].

### 2.5. DNA Extraction from DBSs

DNA was extracted from DBSs using a Chelex-based protocol as optimised and described before [29]. Briefly, one 6 mm DBS punch per sample was soaked overnight in 1 mL of 0.5% (*v*/*v*) Tween 20 (Sigma-Aldrich, Saint-Louis, MO, USA) at 4 °C. The following day, Tween 20 was removed, and the punch underwent a secondary wash in 1 mL PBS (Sigma-Aldrich, Saint-Louis, MO, USA) at 4 °C for 30 min. A 5% (*w*/*v*) Chelex-100 solution (50–100 mesh-size, dry) (Sigma-Aldrich, Saint-Louis, MO, USA) was prepared by dissolving 1 g of resin into 20 mL Tris-EDTA buffer (Sigma-Aldrich, Saint-Louis, MO, USA) and pre-heated to 95 °C. After the removal of the PBS, 50 µL of the prepared 5% Chelex-100 solution was added and heated to 95 °C for 15 min, with a vortexing step every 5 min. Finally, the punches were centrifuged (Eppendorf 5417C, Eppendorf, Hamburg, Germany) for 3 min at maximal speed (11,000× *g*), and the supernatants containing the DNA were collected and stored at −20 °C until further use.

### 2.6. β-Actin, TREC, and KREC Quantification

β-actin (as an internal control), TREC, and KREC copy numbers were quantified using in-house monoplex qPCR. An overview of the primer/probe sequences, master mix composition, and cycling conditions per target is given in Appendix A. The qPCR was run in an end volume of 10 µL, with 2 µL DNA extract and 8 µL Probe Mastermix (Roche Diagnostics, Mannheim, Germany) for the β-actin qPCR and 4 µL DNA extract and 6 µL of Probe Mastermix for the TREC and KREC qPCR. The qPCRs were run on the LightCycler 480 (Roche Diagnostics GmbH, Mannheim, Germany) in duplicate. The mean concentrations of β-actin, TREC, and KREC in copies/mL were calculated from duplicate measurements. Standard series were included using a 1:10 dilution of synthetical DNA fragment; see Appendix A for the sequences (Integrated DNA Technologies, Inc., Coralville, IA, USA).

### 2.7. Statistical Analysis

Samples in which no β-actin was detected were excluded from the statistical analysis. Samples in which no TRECs or KRECs were detected were imputed as 0.01 copies/mL. Next, TREC and KREC concentrations were converted from copies/mL into copies/10^6^ cells to normalise for cell quantity, based on β-actin measurements using the formula in Figure 2 [9]. The normality and lognormality of TREC and KREC copies/10^6^ cells and S1RBD IgG concentrations (IU/mL) were tested and revealed non-normal distributions for TRECs and KRECs and lognormal distribution for S1RBD IgG.

TREC and KREC copies/10^6^ cells were compared between NHRs and NHS, male and female subjects, and infection-naïve and infection-primed subjects by means of a Mann–Whitney test. Simple linear regression was fitted to assess the association between TREC and KREC copies/10^6^ cells and age, log_10_-transformed S1RBD IgG concentrations and age, and TREC and KREC copies/10^6^ cells and log_10_-transformed S1RBD IgG concentrations, with the latter also separately tested for NHRs and NHS. Additionally, two multiple linear regression models were fitted with TREC and KREC copies/10^6^ cells as dependent variables, respectively. The independent variables were a priori selected and based on the literature and the results of the simple linear regression and the Mann–Whitney test: age, sex, infection history, and S1RBD IgG for TRECs and age, sex, and S1RBD IgG for KRECs. Additionally, as a sensitivity analysis, multivariable models were fitted with S1RBD IgG as the dependent variable to assess the explanatory effect of TRECs and KRECs in antibody response. Here, apart from TRECs or KRECs, age, sex, and infection history were included as covariates. To achieve the normality of the residuals for the multiple linear regression model, age and S1RBD were log-transformed, while TREC and KREC levels underwent a Box–Cox transformation with lambda = 0.1 and lambda = 0.4, respectively. *p*-values equal to or below 0.05 were considered statistically significant. All analyses were performed in GraphPad Prism Software Version 10.3.0 (GraphPad Software Inc., San Diego, CA, USA).

## 3. Results

### 3.1. Participant Characteristics

Participant characteristics are given in Table 1. Two subjects (NHS) were excluded from the analysis as no β-actin was detected in their sample. All NHRs and NHS were residing or employed in Belgium. The median age for NHRs and NHS was 86 and 44 years, respectively. The majority of both NHRs and NHS were female. About half of the NHRs and NHS experienced a SARS-CoV-2 infection prior to COVID-19 vaccination.

### 3.2. TREC and KREC Concentrations in NHRs and NHS in Correlation to Age, Sex, and Infection Naivety

TREC (*p* < 0.0001) and KREC levels (*p* < 0.0001) were significantly lower in NHRs compared with NHS (Figure 3). TREC and KREC levels were not detected in 84% (42/50) and 24% (12/50) of NHRs, respectively, and in 14% (21/148) and 6% (9/148) of NHS, respectively. The median TREC level was 0 copies/10^6^ cells (95% CI: 0-0) among NHRs and 1885 copies/10^6^ cells (95% CI: 1290–2290) among NHS. The median KREC level was 1255 copies/10^6^ cells (95% CI: 596–2310) among NHRs and 5420 copies/10^6^ cells (95% CI: 4390–6350) among NHS.

A statistically significant negative correlation was observed between TRECs and age (*p* < 0.0001) and KRECs and age (*p* = 0.0293) in NHRs and NHS combined (Figure 4). The association between age and TRECs (*p* < 0.0001) and age and KRECs (*p* < 0.0001) was confirmed in the multiple linear regression model (Table 2). Furthermore, significantly higher TREC levels (but not KREC levels) were observed among female compared to male participants (*p* = 0.0026) (Appendix A), yet sex was not associated with TREC levels in the multiple linear regression model (*p* = 0.5139). Similarly, TREC levels (but not KREC levels) were higher among infection-primed subjects (*p* = 0.0026) (Appendix A), yet this association was not observed in the multiple linear regression model (*p* = 0.2442).

### 3.3. TREC and KREC Association with SARS-CoV-2 Antibody Response

No statistically significant correlation was found between TREC (*p* = 0.0647) nor KREC levels (*p* = 0.589) and S1RBD IgG concentrations post-primary course vaccination (Figure 5). Also, when NHRs and NHS were considered separately, no correlations were found (*p* > 0.05) (Appendix A). Multiple linear regression also showed no significant association between S1RBD IgG and TREC (*p* = 0.3466) nor KREC levels (*p* = 0.182) (Table 2). Likewise, the sensitivity analysis with S1RBD IgG as the dependent variable showed no explanatory effect of TRECs or KRECs in the SARS-CoV-2 antibody response (Appendix A). Moreover, S1RBD IgG was not statistically significantly correlated to age in our sample (*p* = 0.1317) (Appendix A).

## 4. Discussion

In this study, we examined TRECs and KRECs as markers of thymic and bone marrow output in a population of NHRs and NHS and explored their relationship with SARS-CoV-2 S1RBD-specific IgG antibody responses after COVID-19 vaccination. We found that TREC and KREC levels negatively correlated with age and were therefore statistically significantly lower in NHRs compared to NHS. Simple linear regression indicated that TREC levels were higher in females and individuals with infection-primed immunity; however, these associations were not statistically significant in the multiple linear regression model. Finally, both simple and multiple linear regression showed that S1RBD IgG levels following COVID-19 vaccination were not associated with either TREC or KREC levels.

The findings in our study population confirm the well-documented association between age and TREC levels, reflecting natural thymic involution and the consequent reduction in naïve T-cell output. Indeed, this association has been reported by others, with an even steeper decline in TREC levels observed during childhood compared to adulthood [9,10,30,31]. However, data on TREC levels in older adults are limited, and our findings show that the majority of NHRs had undetectable TREC levels, suggesting a complete absence of naïve T-cell neogenesis. Similarly, we found that KREC levels also decrease with age, indicating the diminished generation of naïve B-cells in the ageing bone marrow. While the literature has similarly reported this negative association between KRECs and age, it is less frequently described in the adult population, as it is more pronounced during paediatric age [9,10,30]. Although no universally accepted thresholds are available for TRECs or KRECs as immunosenescent markers, a study from Kwok et al. quantified TREC and KREC values in different age groups in a healthy Hong Kong population [9]. For comparison, they found a median TREC level of 20,966 copies/10^6^ cells (95% CI: 6644–40,467) and a median KREC level of 22,835 copies/10^6^ cells (95% CI: 7363–103,165) in the age category of 41 to 50 year olds, which aligns with the median age of our NHS population. In their oldest age group (>61 years, with the oldest subject aged 74), they found a median TREC level of 11,668 copies/10^6^ cells (95% CI: 0–35,351) and a median KREC level of 21,092 copies/10^6^ cells (95% CI: 2907–53,665). Compared to the median values in our study population, they found higher TREC and KREC levels, yet their data shows the same age-dependent trend as ours. These differences could be attributed to methodological factors (e.g., different sample type and/or DNA extraction method) or biological differences (e.g., an Asian study population). Likewise, other studies exist characterising TRECs in different adult age categories, yet they report in other units than ours, making a comparison of absolute values difficult [14,32,33,34].

In our study, TREC levels appeared higher in female subjects and infection-primed subjects in the univariate analysis. However, these associations were not retained in the multiple linear regression model, suggesting that age was the predominant determinant of TREC variation in our cohort. Nevertheless, importantly, given the large predominance of female participants in our cohort, sex-stratified interpretations should be made with caution, as this study may have been underpowered for this. In the literature, a study from Lopez et al. also noted higher TREC levels in females [35]. Yet, some studies also demonstrated higher KREC levels among females [9,10], while this was not observed in our cohort.

Additionally, a central aim of our study was to investigate whether TREC and KREC levels were associated with the humoral response following COVID-19 vaccination, functioning as potential biomarkers for vaccine response. Despite the biological plausibility of such a relationship, given the dependence of effective vaccine response on functional naïve T- and B-cells, we found no statistically significant correlation between TREC or KREC levels and S1RBD-specific IgG antibody concentrations. This finding persisted in multivariate models adjusting for age and sex. Two other studies found an association between antibody response after influenza vaccination and TREC levels, one in humans [17] and one in rhesus macaques [18]. A follow-up study from the former paper further explored the association of TRECs with the immune response to influenza vaccination, like cytokine and chemokine expression. They revealed an association between TRECs and miRNAs controlling signalling pathways involved in the activation and differentiation of naïve lymphocytes [13]. Yet, to our knowledge, no other literature exists on the association between TREC or KREC levels and the immune response to vaccination.

Although TREC and KREC levels are associated with lymphocyte neogenesis, vaccine-induced antibody responses involve a complex interplay of additional factors, including pre-existing immunity, antigen presentation, and memory cell recall, which may be more critical in determining humoral response than the new generation of naïve lymphocytes alone. Additionally, it could be hypothesised that, since TRECs and KRECs in the elderly were mostly undetectable, the peripheral clonal expansion of (naïve) T- and B-cells, referred to as homeostatic proliferation, compensates for the lack of lymphocyte neogenesis [36] and ensures the production of an antibody response to COVID-19 vaccination. Indeed, it has been previously described that there are age-related changes in lymphocyte trafficking, and lymphocyte responses in elderly rely on homeostatic proliferation and the peripheral survival of T-and B-cells rather than neogenesis [37,38,39].

Moreover, other components of immune ageing, such as inflammaging, altered cytokine signalling, the role of regulatory T-cells, cellular exhaustion, and impaired germinal centre function, may also contribute to reduced vaccine responsiveness in older adults, independent of thymic and bone marrow output [5,8]. Therefore, while we found no association between TREC or KREC levels and COVID-19 vaccine antibody response, they might be associated with other immune factors, like cellular immunity. Future studies should explore this potential association by means of T-cell and B-cell subset quantification and/or functional assays.

Moreover, while some studies have reported an association between TREC and/or KREC levels and the severity of COVID-19 disease [19,20,21,22], these markers do not appear to be associated with the antibody response to COVID-19 vaccination. As described by Rosichini et al., SARS-CoV-2 can infect the thymus, leading to significant alterations in gene expression profiles and impairing T-cell neogenesis [20]. This thymic disruption may underlie the observed association between reduced TREC levels and increased disease severity. Additionally, the typical cytokine storm during severe COVID-19 might suppress bone marrow haematopoiesis, resulting in reduced KREC levels during acute disease [19,22,40].

This study has limitations, including its cross-sectional design, as both TRECs and KRECs and S1RBD IgG were assessed in the same samples, preventing the assessment of any predictive relationship between TREC or KREC levels and the magnitude or duration of the antibody response. Additionally, TRECs, KRECs, and S1RBD IgG were quantified in samples two months after COVID-19 vaccination, which was the first timepoint available in our study. However, we cannot exclude the possibility that earlier timepoints, when newly generated T-cells and B-cells are more likely to be actively recruited, may show different dynamics, an area that warrants further investigation. Second, this study focused solely on vaccine-induced humoral responses, without evaluating cellular immunity or other immune factors. As TREC and KREC levels were not assessed relative to the proportion of lymphocytes, it is difficult to determine whether low TREC or KREC levels reflect reduced lymphopoiesis or simply lower peripheral lymphocyte numbers. Unfortunately, we could not perform immune phenotyping because of the sample type used (DBS). Lastly, the classification of infection-primed individuals relied on self-reported infection history for NHS and reporting by the NH head nurse, based on the medical files for NHRs. This may have led to an underestimation of the proportion of infection-primed individuals due to recall bias or undetected asymptomatic infections. An assessment of antibodies directed against the SARS-CoV-2 nucleocapsid antigen could aid in estimating the true proportion of individuals who previously experienced a SARS-CoV-2 infection; however this is also not completely reliable as antibodies tend to become undetectable within a year after infection [41].

## 5. Conclusions

The levels of TRECs and KRECs, markers of recent thymic and bone marrow output, respectively, were statistically significantly lower in NHRs compared with NHS, reflecting their association with age. We found no association between TREC or KREC levels and the SARS-CoV-2 S1RBD-specific IgG antibody responses following COVID-19 vaccination. These findings suggest that while TREC and KREC levels are markers of lymphocyte neogenesis and immunological ageing, they may not independently predict humoral vaccine responses in (older) adults. Further research is warranted to explore their potential relevance in cellular immune responses or in combination with other biomarkers of immune function.

## Figures and Tables

**Figure 1 vaccines-13-00874-f001:**
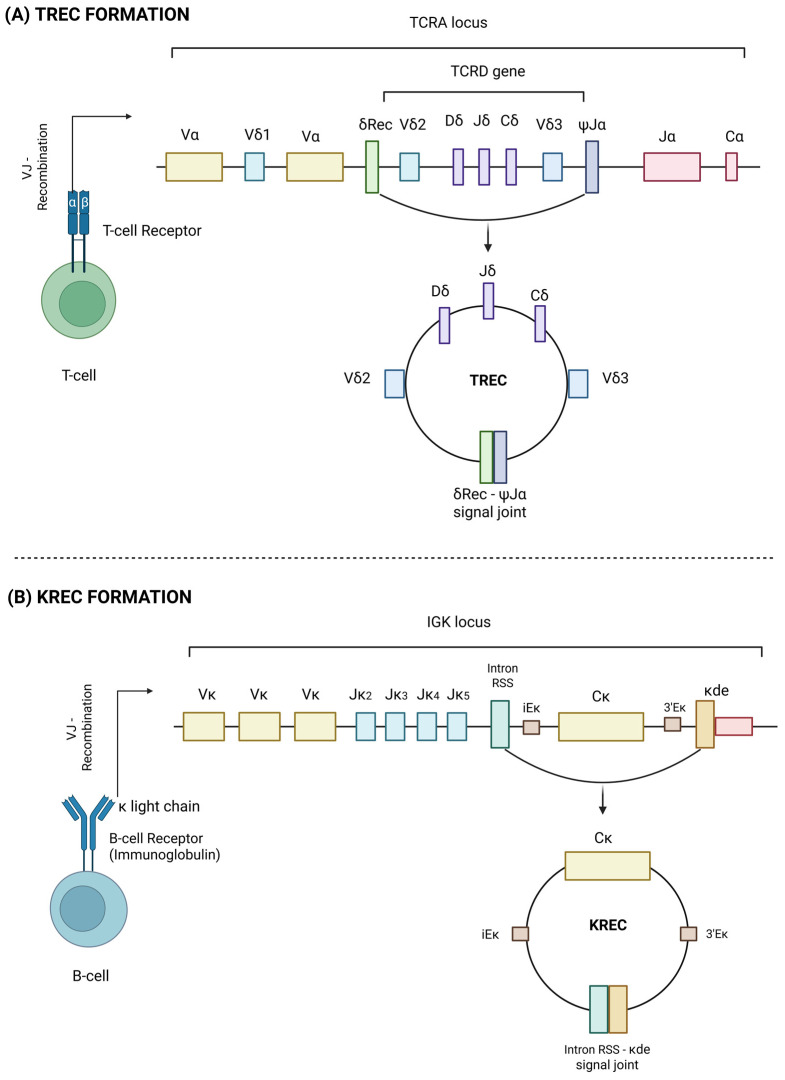
A schematic overview of the formation of (**A**) T-cell receptor excision circles (TRECs) and (**B**) kappa-deleting recombination excision circles (KRECs) during T-cell and B-cell maturation, respectively.

**Figure 2 vaccines-13-00874-f002:**
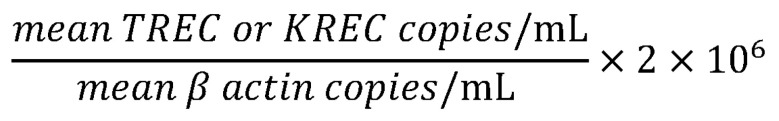
Formula to convert TREC or KREC copies/mL to copies/10^6^ cells [9].

**Figure 3 vaccines-13-00874-f003:**
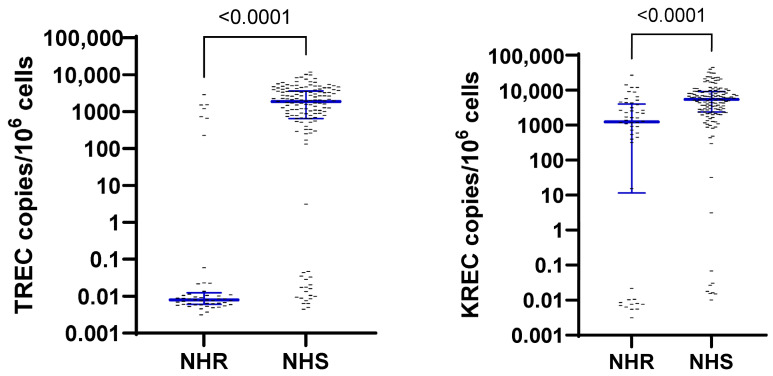
T-cell receptor excision circles (TRECs) and kappa-deleting recombination excision circles (KRECs) per 10^6^ cells in NHRs (n = 50) and NHS (n = 148). Blue horizontal lines represent median value with interquartile range.

**Figure 4 vaccines-13-00874-f004:**
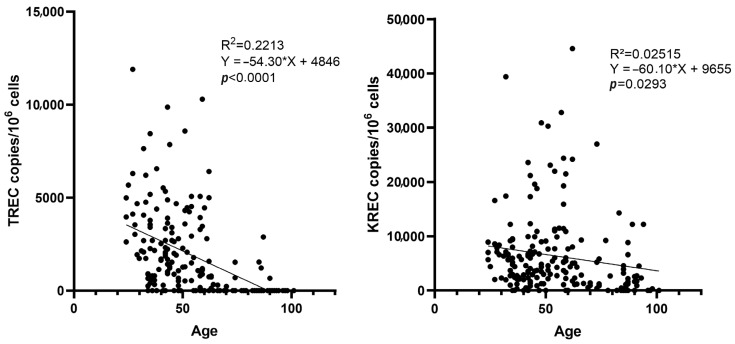
Simple linear regression between T-cell receptor excision circles (TRECs) (**left**) and kappa-deleting recombination excision circles (KRECs) (**right**) and age in NHRs (n = 50) and NHS (n = 148). The correlation coefficient (R^2^), the equation representing the relationship between TREC or KREC levels (Y) and age (X), and *p*-value are shown in this figure. A *p*-value ≤ 0.05 is considered statistically significant.

**Figure 5 vaccines-13-00874-f005:**
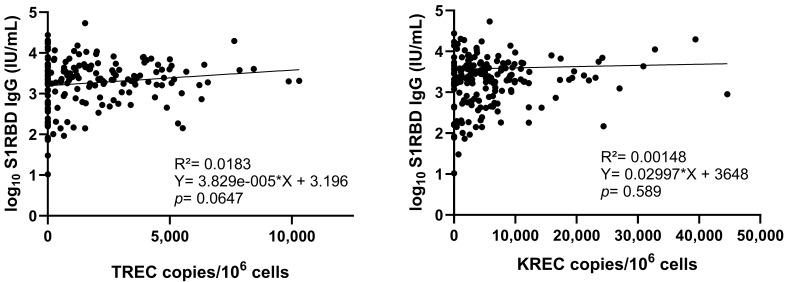
Scatterplots between T-cell receptor excision circle (TREC) (**left**) and kappa-deleting recombination excision circle (KREC) (**right**) levels and S1RBD IgG antibody concentrations after COVID-19 vaccination in NHRs (n = 50) and NHS (n = 148). International units/mL: IU/mL. The correlation coefficient (R^2^), the equation representing the relationship between log10 S1RBD IgG levels (Y) and TREC or KREC levels (X), and *p*-value are shown in this figure. A *p*-value ≤0.05 is considered statistically significant.

**Table 1 vaccines-13-00874-t001:** Participant characteristics. NHR, nursing home resident; NHS, nursing home staff; IQR, interquartile range; n, number. NA, not applicable.

	NHR (n = 50)	NHS (n = 148)	NHR + NHS (n = 198)
Age, median (IQR)	86 (77–90)	44 (36–54)	51 (40–66)
Female sex, n (%)	35 (70)	130 (88)	165 (83)
Infection-primed, n (%)	30 (60)	78 (53)	98 (49)
Cardiovascular disease, n (%)	14 (28)	0 (0)	14 (7)
Diabetes, n (%)	7 (14)	0 (0)	7 (4)
Hypertension, n (%)	11 (22)	0 (0)	11 (6)
Severe renal, lung, or heart disease, n (%)	2 (4)	0 (0)	2 (1)
Immunodeficiency and/ or immunosuppression, n (%)	1 (2)	0 (0)	1 (1)
Active cancer, n (%)	1 (2)	0 (0)	1 (1)
Highly care-dependent, n (%) (Katz dependency scale C, Cd)	17 (34)	NA	NA
Little to no care dependence, n (%) (Katz dependency scale O, A, B)	32 (64)	NA	NA

**Table 2 vaccines-13-00874-t002:** Multiple linear regression models for TRECs (T-cell receptor excision circles) and KRECs (kappa-deleting recombination excision circles) in NHRs (n = 50) and NHS (n = 148). Statistically significant *p*-values are underlined.

	Variable	Estimate	Standard Error	95% Confidence Interval	*p* Value
TREC ^b^	Intercept	51.82	5.689	40.60 to 63.05	<0.0001
	Age (years) ^a^	−28.69	2.692	−34.00 to −23.38	<0.0001
	S1RBD IgG concentration ^a^	0.7574	0.8026	−0.8264 to 2.341	0.3466
	Female	0.7648	1.169	−1.543 to 3.072	0.5139
	Infection-primed	1.105	0.9458	−0.7612 to 2.971	0.2442
KREC ^b^	Intercept	81.03	19.26	43.02 to 119.0	<0.0001
	Age (years) ^a^	−0.5440	0.1386	−0.8174 to −0.2706	0.0001
	S1RBD IgG concentration ^a^	6.166	4.603	−2.917 to 15.25	0.1821
	Female	−6.231	7.587	−21.20 to 8.739	0.4125

^a^ Age and S1RBD were log_10_-transformed; ^b^ TREC and KREC levels underwent a Box–Cox transformation with lambda = 0.1 and lambda = 0.4, respectively.

## Data Availability

The data are available upon request.

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
