# Peer review of "Exploring TREC and KREC Levels in Nursing Home Residents and Staff and Their Association with SARS-CoV-2 Antibody Response After Vaccination"

_vaccines, 2025, doi:10.3390/vaccines13080874_

Round 1

Reviewer 1 Report

Comments and Suggestions for Authors

The paper compares the levels of TREC and KREC in relation with the titer of anti RBD antibodies two months after SARS Cov2 vaccination, in 2 cohorts of subjects of different age.

The results indicate that the number of TREC and KREC decreases with age but are not associated with the levels of anti RBD antibodies.

The paper investigates a relevant issue in the analysis of the impact of age on the immune response to vaccines by means of reliable techniques.

However, the analysis of TREC and KREC at a single time point raises some concern.

Can the authors claim that naïve T and B cells are still recruited in the immune response to RBD two months after vaccination?

Can they obtain data at multiple time points?

How is the titer of anti RBD antibodies related with age?

Author Response

The paper compares the levels of TREC and KREC in relation with the titer of anti RBD antibodies two months after SARS Cov2 vaccination, in 2 cohorts of subjects of different age. The results indicate that the number of TREC and KREC decreases with age but are not associated with the levels of anti RBD antibodies. The paper investigates a relevant issue in the analysis of the impact of age on the immune response to vaccines by means of reliable techniques.

  • However, the analysis of TREC and KREC at a single time point raises some concern. Can the authors claim that naïve T and B cells are still recruited in the immune response to RBD two months after vaccination? Can they obtain data at multiple time points?

We would like to thank the reviewer for this interesting comment. We agree that we cannot claim that newly generated naïve T and B cells are actively recruited into the immune response two months after vaccination, as the peak of naïve cell recruitment likely occurs earlier in the immune response. However, it is important to emphasize that TRECs and KRECs are not antigen-specific and therefore do not reflect whether lymphocytes are specifically recruited into the immune response against SARS-CoV-2 antigens. Nevertheless, immunological challenges such as infection or vaccination might impact the level of TRECs and KRECs. For example, a recent study demonstrated that the TREC and KREC levels were significantly lower in the acute stage of COVID-19 than in the convalescent phase (1).

Our hypothesis was that reduced lymphocyte neogenesis might be associated with a weaker antibody response after vaccination. However, our data did not support this assumption. As discussed in the manuscript, other mechanisms such as peripheral homeostatic proliferation, may compensate for reduced production of new lymphocytes (line 331-337).

Nevertheless, we should acknowledge the limitation that TREC and KREC levels were measured two months post-vaccination, as it remains uncertain whether measurements taken at earlier time points (e.g., 14 days post-vaccination) would yield different results. Unfortunately, we have no samples available earlier than two months after COVID-19 vaccination. Indeed, current knowledge about the temporal dynamics of TREC and KREC levels (following vaccination) is limited. Future studies should aim to investigate the dynamics of TRECs and KRECs for better understanding of these markers.

In the revised manuscript, we have discussed the cross-sectional design and timepoint of measuring as a limitation in line 355-362.

Reference:

  1. Savchenko AA, Tikhonova E, Kudryavtsev I, Kudlay D, Korsunsky I, Beleniuk V, et al. TREC/KREC Levels and T and B Lymphocyte Subpopulations in COVID-19 Patients at Different Stages of the Disease. Viruses [Internet]. 2022; 14(3).

  • How is the titer of anti RBD antibodies related with age?

We have previously evaluated this association to evaluate co-linearity in the multiple regression model, but this data was not included in the previous version of our manuscript. In our sample, age was not correlated to S1RBD IgG levels (R²= 0.0123; p=0.1317). In the revised version of our manuscript, we have included this figure in the supplementary file (Supplementary Figure 3; manuscript line 262).

Reviewer 2 Report

Comments and Suggestions for Authors

Thank so much for your consideration to me to review this manuscript. I think it's a worthy, well-structured and interesting article. This manuscript represents an interesting research regarding T-cell receptor excision circles and kappa-deleting recombination 21 excision circles and their association to anti-spike SARS-CoV-2 (S1RBD) IgG antibody response after COVID-19 vaccination.

I don’t have find any special problems. Limitations are well described and justified.

Nice study. It can be accepted as it.

Author Response

Thank so much for your consideration to me to review this manuscript. I think it's a worthy, well-structured and interesting article. This manuscript represents an interesting research regarding T-cell receptor excision circles and kappa-deleting recombination 21 excision circles and their association to anti-spike SARS-CoV-2 (S1RBD) IgG antibody response after COVID-19 vaccination.

I don’t have find any special problems. Limitations are well described and justified.

Nice study. It can be accepted as it.

We would like to thank the reviewer for their time to review our manuscript and their positive feedback.

Reviewer 3 Report

Comments and Suggestions for Authors

This cross-sectional study investigates the association between thymic (TREC) and B-cell (KREC) outcomes and the humoral response (anti-S1RBD IgG) to COVID-19 vaccination in a cohort of 50 nursing home residents (NHR) and 150 staff (NHS) by using dried blood spots (DBS) as the sampling method. The study is methodologically sound and presents valuable data on ageing-related immunological dynamics. 

Commens.
1. Inconsistent English style (UK vs US spelling):
The manuscript inconsistently uses UK and US spelling conventions (e.g., paediatric vs. hematopoiesis). Consider revising for consistency throughout the manuscript. Either UK or US English may be used, but it must be applied uniformly.

2. Clinical relevance of TREC and KREC values:
Although TREC and KREC levels are central to the study. The manuscript lacks information on clinical reference ranges, interpretive thresholds, or how the observed values relate to established immunosenescence markers. 
A brief contextualisation in the Introduction or Discussion would be helpful for readers less familiar with these assays.

3. Potential for misclassification bias (Infection Status):
Prior SARS-CoV-2 infection status was based on self-reporting. Given the risk of asymptomatic infections, especially in healthcare settings. This could introduce the misclassification bias. Consider acknowledging this limitation more clearly in the Discussion, and comment on how serological (anti-N) data or clinical records could be used in future studies to validate exposure status.

4. Absence of cellular immune profiling:
The manuscript assesses only humoral immunity (anti-S1RBD IgG), despite measuring thymic and B-cell neogenesis markers (TREC and KREC), which are more directly related to cellular immunity. Even if not measured in this study. It would be worthwhile to recommend that future research incorporate T- and B-cell subset quantification or functional assays (e.g., AIM or ICS) to complement TREC and KREC analysis.

5. Lack of association between TREC, KREC and humoral immunity:
The manuscript reports no significant association between TREC, KREC levels and anti-S1RBD antibody titres. This is a noteworthy finding that should be emphasised more clearly in the Discussion section. As TRECs and KRECs are indicators of recent thymic and bone marrow output (reflecting T- and B-cell neogenesis, respectively). Their lack of correlation with humoral response raises important biological questions.

Consider expanding the discussion to include plausible immunological explanations, such as:
-Waning adaptive immune memory with ageing
-Altered lymphocyte trafficking or peripheral homeostasis in senescence
-Functional dysregulation of T- and B-cell responses rather than numerical deficits
-The role of regulatory T-cells or exhaustion markers in shaping immune responses in elderly or previously infected individuals

Furthermore, the male subgroup in both nursing home residents (NHR) and staff (NHS) was relatively small, which may lead to underpowered comparisons and an inability to detect statistically significant sex-based differences in TREC and KREC levels (e.g., the lack of significance in KREC comparisons between males and females). 
Suggest acknowledging this limitation and its potential implications for sex-stratified interpretations.

Typos.
1. Instrument name:
Use correct naming: "LightCycler 480" (with a space), as per the manufacturer's standard.
See: https://diagnostics.roche.com/global/en/products/instruments/lightcycler-480-ins-445.html

Author Response

This cross-sectional study investigates the association between thymic (TREC) and B-cell (KREC) outcomes and the humoral response (anti-S1RBD IgG) to COVID-19 vaccination in a cohort of 50 nursing home residents (NHR) and 150 staff (NHS) by using dried blood spots (DBS) as the sampling method. The study is methodologically sound and presents valuable data on ageing-related immunological dynamics. 

We thank the reviewer for their time to review and suggestions. Please find our rebuttal below.

Comments.
1. Inconsistent English style (UK vs US spelling):
The manuscript inconsistently uses UK and US spelling conventions (e.g., paediatric vs. hematopoiesis). Consider revising for consistency throughout the manuscript. Either UK or US English may be used, but it must be applied uniformly.

We thank the reviewer for this thoughtful remark. We have revised the manuscript according to British spelling.

  1. Clinical relevance of TREC and KREC values:
    Although TREC and KREC levels are central to the study. The manuscript lacks information on clinical reference ranges, interpretive thresholds, or how the observed values relate to established immunosenescence markers. 
    A brief contextualisation in the Introduction or Discussion would be helpful for readers less familiar with these assays.

We thank the reviewer for this suggestion. Although there are no thresholds described for TRECS or KRECS as immunosenescent markers, there is a study in a Hong Kong population that describes median TREC and KREC ranges in different age groups, which can be insightful for the interpretation of our results. We have therefore added this to the discussion in line 290-302.

  1. Potential for misclassification bias (Infection Status):
    Prior SARS-CoV-2 infection status was based on self-reporting. Given the risk of asymptomatic infections, especially in healthcare settings. This could introduce the misclassification bias. Consider acknowledging this limitation more clearly in the Discussion, and comment on how serological (anti-N) data or clinical records could be used in future studies to validate exposure status.

We have elaborated on this matter in the limitation section line 367-374. We would like to note that for nursing home residents, this information was based on their medical file, and we have clarified this in the manuscript. Moreover, while measuring anti-N antibodies might indeed give biological proof of a previous SARS-CoV-2 infection, studies have shown that they tend to become undetectable within a year after infection, and are therefore not 100% reliable.

  1. Absence of cellular immune profiling:
    The manuscript assesses only humoral immunity (anti-S1RBD IgG), despite measuring thymic and B-cell neogenesis markers (TREC and KREC), which are more directly related to cellular immunity. Even if not measured in this study. It would be worthwhile to recommend that future research incorporate T- and B-cell subset quantification or functional assays (e.g., AIM or ICS) to complement TREC and KREC analysis.

We acknowledge this limitation and it is included in the manuscript as part of the conclusion. Moreover we have added a more specific recommendation of what future research should entail to explore the association between TREC/KRECS and cellular immunity in line 340-345.

  1. Lack of association between TREC, KREC and humoral immunity:
    The manuscript reports no significant association between TREC, KREC levels and anti-S1RBD antibody titres. This is a noteworthy finding that should be emphasised more clearly in the Discussion section. As TRECs and KRECs are indicators of recent thymic and bone marrow output (reflecting T- and B-cell neogenesis, respectively). Their lack of correlation with humoral response raises important biological questions. Consider expanding the discussion to include plausible immunological explanations, such as:
    -Waning adaptive immune memory with ageing
    -Altered lymphocyte trafficking or peripheral homeostasis in senescence
    -Functional dysregulation of T- and B-cell responses rather than numerical deficits
    -The role of regulatory T-cells or exhaustion markers in shaping immune responses in elderly or previously infected individuals

We agree with the reviewer that this was a central aim in our study, and our findings are therefore clearly emphasized in the discussion from line 313 to 345. In the revision of our manuscript, we have added a more detailed discussion on the concept of peripheral homeostatic proliferation, which might indeed be a plausible immunological explanation for the lack of association between TRECS/KRECS and COVID-19 vaccine antibody response (line 330-340).

Furthermore, the male subgroup in both nursing home residents (NHR) and staff (NHS) was relatively small, which may lead to underpowered comparisons and an inability to detect statistically significant sex-based differences in TREC and KREC levels (e.g., the lack of significance in KREC comparisons between males and females). 
Suggest acknowledging this limitation and its potential implications for sex-stratified interpretations.

We indeed acknowledge this limitation, it is discussed in the manuscript in line 306-308.

Typos.
1. Instrument name:
Use correct naming: "LightCycler 480" (with a space), as per the manufacturer's standard.
See: https://diagnostics.roche.com/global/en/products/instruments/lightcycler-480-ins-445.html

This was corrected.

Reviewer 4 Report

Comments and Suggestions for Authors

My comments are as follows:

  1. As a statistical conclusion depends largely on the enrolled population, the authors should provide more information of the participants, for example, the nationality, the immune background, et al. And how the sample size was determined should be cleared stated.
  2. While the author did not found any correlation between TREC/KREC levels and antibody response, and disturbance of TREC/KREC levels often indicates immunodeficiency, I recommended further investigating the relationship between TREC/KREC levels and peripheral immune cell composition.

Author Response

My comments are as follows:

  1. As a statistical conclusion depends largely on the enrolled population, the authors should provide more information of the participants, for example, the nationality, the immune background, et al. And how the sample size was determined should be cleared stated.

We thank the reviewer for this thoughtful comment. We have included additional information on the population (line 220), including the prevalence of comorbidities, in Table 1 of the revised manuscript. Additionally, in line 127-134 of the revised manuscript, we aimed to clarify the sample size calculation.

  1. While the author did not found any correlation between TREC/KREC levels and antibody response, and disturbance of TREC/KREC levels often indicates immunodeficiency, I recommended further investigating the relationship between TREC/KREC levels and peripheral immune cell composition.

We acknowledge the limitation of not assessing cellular immune cell composition. However, given that we used Dried Blood Spots as a sample type, unfortunately, we are not able to perform assessment of cellular immune responses like immune phenotyping by flow cytometry or functional immunoassays like ELISpot. We have included this as a limitation (line 358-368) and additionally discussed this in the discussion of the revised manuscript in line 338-344.

Round 2

Reviewer 1 Report

Comments and Suggestions for Authors

No further modification is required.